# Regulation and Liquid Sensing of Electromagnetically Induced Transparency-like Phenomena Implemented in a SNAP Microresonator

**DOI:** 10.3390/s24217069

**Published:** 2024-11-02

**Authors:** Chenxiang Liu, Minggang Chai, Chenglong Zheng, Chengfeng Xie, Chuanming Sun, Jiulin Shi, Xingdao He, Mengyu Wang

**Affiliations:** Key Laboratory for Optoelectronic Information Perception and Instrumentation of Jiangxi, Nanchang Hangkong University, Nanchang 330063, China; 2208080300006@stu.nchu.edu.cn (C.L.); zcl15248664086@163.com (C.Z.); xcf@nchu.edu.cn (C.X.); ih2112scm@163.com (C.S.); jlshi@nchu.edu.cn (J.S.); hxd@nchu.edu.cn (X.H.)

**Keywords:** electromagnetically induced transparency-like, double-EIT-like effect, multi-mode interference, liquid sensing, displacement sensing

## Abstract

Optical microresonators supporting whispering-gallery modes (WGMs) have become a versatile platform for achieving electromagnetically induced transparency-like (EIT-like) phenomena. We theoretically and experimentally demonstrated the tunable coupled-mode induced transparency based on the surface nanoscale axial photonics (SNAP) microresonator. Single-EIT-like and double-EIT-like (DEIT-like) effects with one or more transparent windows are achieved due to dense mode families and tunable resonant frequencies. The experimental results can be well-fitted by the coupled mode theory. An automatically adjustable EIT-like effect is discovered by immersing the sensing region of the SNAP microresonator into an aqueous environment. The sharp lineshape and high slope of the transparent window allow us to achieve a liquid refractive index sensitivity of 2058.8 pm/RIU. Furthermore, we investigated a displacement sensing phenomenon by monitoring changes in the slope of the transparent window. We believe that the above results pave the way for multi-channel all-optical switching devices, multi-channel optical communications, and biochemical sensing processing.

## 1. Introduction

In the past decades, electromagnetically induced transparency (EIT) has attracted great research interests due to its distinct properties and wide applications in slow light propagation, quantum information, nonlinearity enhancement, optical sensing, and optical communication. The EIT effect was first observed in atomic media [1]. Right now, EIT-like tuning phenomenon can be found in coupled-cavity systems [2,3,4,5]. WGM microresonators have emerged as excellent candidates for achieving a conspicuous EIT-like spectrum owing to their high-quality (*Q*) factors and small mode volume. EIT-like effects have been generated and regulated in dual-waveguide configuration [6], multi-microcavity coupling configuration [7], and single microcavity configuration [8]. Realization of an EIT-like effect in a single microcavity greatly decreases experimental complexity, compared with multi-microcavity coupling systems and dual-waveguide configurations. In recent years, some novel approaches have been developed to tune EIT-like lineshapes, such as temperature tuning [9], pressure tuning [10], strain mechanical tuning [11], and coupled gap tuning [12]. For these systems, tuning methods and additional technologies are more complicated and expensive. Therefore, it is desirable to develop an efficient and simple method to realize EIT-like lineshapes using a single microcavity. Utilizing the slope of the transparent window to record environmental changes is a novel sensing method, especially suitable for microresonators with low *Q* factors. Many works have proven that the high slope of the Fano lineshape is more suitable for capturing tiny signals at a fixed resonant wavelength, compared with that of the Lorentz lineshape [13,14,15]. Ulteriorly, high detection sensitivity can be achieved profit from the sharp lineshape of the transparent window.

Given the practical requirements for EIT-like regulation, we focus on using the SNAP microresonator as an ideal candidate to accomplish this work. The SNAP microresonator has only a nanometer-scale bump on its surface, and the axial mode field extends along the longitudinal axis of the microresonator [16,17,18,19]. The microresonator can support rich mode families, and the tunability of the resonant frequency enables convenient regulation for EIT-like effects based on dynamic mode-coupling theory. More importantly, it is an ideal platform for achieving axial separation sensing. Thanks to the axial mode field distribution extending along the longitudinal axis, we can select the sensing region in the axial direction rather than just in the coupling plane. The spatial separation between the coupling region and the sensing region not only prevents analyte disturbance on the coupling region but also ensures stable WGMs excitation. Compared to immersion systems [20,21], the vulnerable tapered fibers can be effectively protected using this separation scheme. Due to the superior anti-interference performance reflected in this scheme, the output signal is stable and accurate.

In this paper, we demonstrate the tunable EIT-like effect using only a single SNAP microresonator by selecting the microresonator–taper coupling plane along the longitudinal axis vertically. In particular, we investigated the DEIT-like lineshapes with two transparent windows. Using the coupled-mode theory, these experimental lineshapes can be well-fitted. Subsequently, the sensing region of the SNAP microresonator was immersed in liquid (loaded into a microfluidic channel) to evaluate the axial separation sensing performance. Interestingly, an automatically adjusted EIT-like effect appears in this detection system. By changing the liquid refractive index and utilizing the sharp lineshape of the transparent window, we obtained a liquid refractive index sensitivity (RIS) of 2058.8 pm/RIU. In addition, we investigated displacement sensing using the high slope of the transparent window, achieved by continuously decreasing the distance L between the microresonator–taper coupling plane and the liquid surface. This phenomenon can be applied to displacement detectors. Implementing an EIT-like effect in the SNAP microresonator and using it for sensing has enormous potential in multi-channel optical communications, all-optical switching devices, and ultra-sensitive biochemical detection.

## 2. Theoretical Model

SNAP microresonators are prepared using the arc discharge method [22]. A microcapillary with a removed coating layer is fixed by two V-groove clamps on the fusion splicer (Fujikura Corporation, Tokyo, Japan FSM-40S), while controlling the fusion splicer discharge. This operation induces a protrusion ~100 nm on the outer wall of the microcapillary. Given that the length of the molten region is 400 μm, the curvature of the SNAP microresonator can be calculated to be 0.00028 μm^−1^. The initial radius of the microcapillary is 62.5 µm. The electric field distribution can be obtained by substituting these parameters into the simulation model, as shown in Figure 1. The resonant modes are characterized by three indices: the radial mode number (*p*), the azimuth mode number (*m*), and the axial mode number (*q*). These can be expressed together in the form (*p*, *m*, *q*). The simulation model mentioned here was established in MATLAB R2021a, and the two-dimensional axisymmetric morphology was determined by the parameters mentioned above. We can present a full-scale wave equation for the SNAP microresonator to theoretically investigate the WGM properties. The theoretical model here is compatible with the model constructed for the bottle microresonator [23]. In the orthogonal curvilinear coordinates (z, r, φ), the optical field distribution in the bottle microresonator is given by the separated wave equation [24]. The mode field distribution of the SNAP microresonator can be calculated as $E_{\left(p,m,q\right)}\left(r,z\right)=E_{p}\left(r,z\right)\cdot Z_{m,q}\left(z\right)\cdot e^{im\varphi }$, where $E_{p}\left(r,z\right)$ is the radial field distribution, $Z_{m,q}\left(z\right)$ is the axial field distribution, and $e^{im\varphi }$ is the azimuth component. The resonant wavelength $\lambda_{\left(p,m,q\right)}$ for each mode is given by [25]:(1)$$\lambda_{\left(p,m,q\right)}=2\pi n_{eff}{\left[{\left(\frac{m}{c_{r}R_{0}}\right)}^{2}+\left(q+\frac{1}{2}\right)\Delta E_{m}\right]}^{-\frac{1}{2}}$$
where $\Delta E_{m}=2m∆k/c_{r}.R_{0}$ denotes the maximum radius of the bottle region. $c_{r}$ is the correction factor, which can be obtained from the radial field distribution with *q* = 0.

Figure 1a illustrates the normalized electric field distribution for the fifth order axial mode. It is worth noting that the axial mode field distribution reaches ±200 μm along the z-coordinate, which means that a farther evanescent field formed in the axial direction which is beneficial for axial separation sensing. The normalized electric field distribution for the third order radial mode (*p* = 3) and the cross-sectional view with (*p* = 1, *m* = 365, *q* = 0), (*p* = 2, *m* = 337, *q* = 2), and (*p* = 3, *m* = 319, *q* = 4) can also be seen in Figure 1b,c. Figure 1d shows the resonant wavelengths as a function of the axial mode numbers with measurement size. It can be seen that, for example, the resonant wavelength of mode (3, 319, *q*) is localized between modes (1, 365, *q*) and (2, 337, *q*). The resonant wavelength point of modes (1, 365, *q*) and (3, 319, *q*) is closely adjacent at the same *q*. In fact, there will be more adjacent and overlapping modes if we consider more mode combinations. Dense mode families have overlapping frequency distribution zones, which makes it possible to realize mode-coupling tuning [11]. On the other hand, the axial effective length (longitudinal axis direction) of the protrusion with nanometer scale can reach several hundred microns. This property enables more possibilities for both single-EIT-like effect and DEIT–like effect based on multi-mode-coupling theory.

Different WGMs excited in the SNAP microresonator have different tuning rates [23], and mode-coupling can be controlled. Figure 2 shows a schematic diagram of the mode-coupling effect for three resonant modes. In Figure 2c, Mode *A* and Mode 2 are in a zero-detuning state, thus exhibiting a transparent window. However, the resonant frequency of Mode 3 is different from Mode *A* (2) and exhibits a Fano lineshape in the spectrum. For the resonant modes Mode *A* and Mode *i*, without ignoring their phase changes, coupled-mode equations for the microresonator–taper system can be expressed:(2)$$\frac{d\alpha_{A}}{dt}=-i\Delta \omega_{A}\alpha_{A}-\frac{\gamma_{A}+\kappa_{A}}{2}\alpha_{A}-\sum_{i}g_{Ai}\alpha_{i}+\sqrt{\kappa_{A}}S_{\mathrm{in}}\cos\varphi_{A}$$
(3)$$\frac{d\alpha_{i}}{dt}=-i\Delta \omega_{i}\alpha_{i}-\frac{\gamma_{i}+\kappa_{i}}{2}\alpha_{i}-g_{Ai}\alpha_{A}+\sqrt{\kappa_{i}}S_{\mathrm{in}}\cos\varphi_{i}$$
where $S_{\mathrm{in}}$ represents the incident light field amplitude. $\alpha_{A}$ denotes the lower-*Q* mode termed valley mode and $\alpha_{i}$ represents the higher-*Q* mode, which denotes the transparency window and absorption dip, respectively. $\omega_{A/i}$ are the resonant frequencies for the corresponding modes, and ${∆\omega }_{A/i}=\omega -\omega_{A/i}$, ω is the frequency detuning. $\gamma_{A/i}$ denotes the intrinsic losses of Mode *A* and Mode *i*. $\kappa_{A/i}$ reperents the external coupling losses of Mode *A* and Mode *i*. $\varphi_{A/i}$ is the polarization angle of Mode *A* and Mode *i* relative to $S_{\mathrm{in}}$. $g_{Ai}$ is the coupling strengths between Mode *A* and Mode *i*, $g_{Ai}=\left(\sqrt{\kappa_{A}\kappa_{i}}\cos\left(\varphi_{i}+\varphi_{A}\right)\right)/2$. It should be noted that the above time equation is derived from the Hamiltonian description of the microresonator–taper system [26].

Input/output relationships for the two orthogonal polarizations (horizontal $S_{\mathrm{out}\;}^{h}$ and vertical $S_{\mathrm{out}\;}^{h}$) are expressed as follows:(4)$$S_{\mathrm{out}\;}^{h}=-S_{\mathrm{in}}\cos\varphi_{A}+\sqrt{\kappa_{A}}\alpha_{A}+\sum_{i}\sqrt{\kappa_{i}}\alpha_{i}\cos\left(\varphi_{i}+\varphi_{A}\right)$$
(5)$$S_{\mathrm{out}\;}^{v}=-S_{\mathrm{in}}\sin\varphi_{A}+\sum_{i}\sqrt{\kappa_{i}}\alpha_{i}\sin\left(\varphi_{i}+\varphi_{A}\right)$$

The output light field can be expressed as: ${\left|S_{\mathrm{out}\;}\right|}^{2}={\left|S_{\mathrm{out}\;}^{h}\right|}^{2}+{\left|S_{\mathrm{out}\;}^{\nu }\right|}^{2}$. Based on this, the normalized transmission spectrum is represented as:(6)$$\begin{array}{c} T_{\mathrm{cavity}\;}=\frac{{\left|S_{\mathrm{out}\;}\right|}^{2}}{{\left|S_{\mathrm{in}\;}\right|}^{2}}=\frac{1}{{\left|S_{\mathrm{in}\;}\right|}^{2}}\left({\left|-S_{\mathrm{in}\;}\cos\varphi_{A}+\sqrt{\kappa_{A}}\alpha_{A}+\sum_{i}\sqrt{\kappa_{i}}\alpha_{i}\cos\left(\varphi_{i}+\varphi_{A}\right)\right|}^{2}\right.\\ \left.+{\left|-S_{\mathrm{in}\;}\sin\varphi_{A}+\sum_{i}\sqrt{\kappa_{i}}\alpha_{i}\sin\left(\varphi_{i}+\varphi_{A}\right)\right|}^{2}\right) \end{array}$$

The EIT-like/Fano effects and the corresponding multiple transparent windows can be achieved through tuning $\varphi_{A/i}$, $\gamma_{A/i}$, $\kappa_{A/i}$, $g_{Ai}$, and $\Delta \omega_{A/i}$. By moving the SNAP microresonator vertically along its longitudinal axis, the entire coupling system goes through different coupling regions, allowing for the tuning of $\kappa_{A/i}$ and $\Delta \omega_{A/i}$. In this paper, the units for the above parameters are set to: $\gamma_{A/i}$ (MHz), $\kappa_{A/i}$ (MHz), $\omega_{A/i}$ (MHz), $g_{Ai}$ (MHz), and $\varphi_{A/i}$ (Rad).

For WGM-based microcavity detection systems, sensitivity is typically presented by spectral deformation or shift, which can be expressed as:(7)$$\frac{dT_{\mathrm{cavity}\;}}{dn_{s}}=\frac{dT_{\mathrm{cavity}\;}}{d\lambda }\frac{d\lambda }{dn_{s}}=\propto \frac{dT_{\mathrm{cavity}\;}}{d\omega }$$
where $n_{s}$ denotes the effective refractive index and λ denotes the resonant wavelength. In addition to the resonant wavelength shift, detection sensitivity can also be observed directly in the slope of the spectral lineshape at a fixed wavelength. Here, the detection sensitivity is represented as $S_{lope}\equiv \left|dT_{\mathrm{cavity}\;}/d\omega \right|$. This method is particularly suitable for amplifying tiny sensing signals, but a high slope lineshape is required. An EIT-like effect will form a high slope within the transparent window, facilitating precision sensing at a fixed wavelength.

Figure 3a shows the simulation spectra without a transparent window between the high-*Q* mode and the low-*Q* mode, with the fitting parameter set to $\left[\gamma_{1},\gamma_{2},\kappa_{1},\kappa_{2},\omega_{1},\omega_{2},\varphi_{1},\varphi_{2}\right]=\left[50\pi \sim 80\pi ,2\pi ,10\pi ,0,-80\pi ,0,0.71,4.28\right]$. $\gamma_{1/2},\kappa_{1/2},\omega_{1/2},$ and $\varphi_{1/2}$ are the fitting data for the low-*Q* mode and the high-*Q* mode, respectively. To obtain a more general spectral evolution, we set a range of 50π to 80π at $\gamma_{1}$. By substituting the above fitting parameters into Equation (6), the simulation spectra can be obtained. On the other hand, setting the fitting parameter to $\left[\gamma_{1},\gamma_{2},\kappa_{1},\kappa_{2},\omega_{1},\omega_{2},\varphi_{1},\varphi_{2}\right]=\left[50\pi \sim 80\pi ,2\pi ,10\pi ,0,-80\pi ,-80\pi ,0.71,4.28\right]$ can result in a standard transparent window, where the resonant frequency difference between high-*Q* and low-*Q* modes is zero, as shown in Figure 3b. On the basis of Figure 3a,b, when we take the derivative of $T_{\mathrm{cavity}\;}$ with ω in Equation (6) to obtain the spectra slope distribution shown in Figure 3c,d, the relevant parameters remain unchanged. The spectra with $\gamma_{1}$ = 150 were extracted from Figure 3a,b and are presented in Figure 3e. Concurrently, their slope curves are shown in Figure 3f (i.e., the slope curves with $\gamma_{1}$ = 150 in Figure 3c,d). As shown in Figure 3e, Fano resonance will be displayed as the resonant frequency difference of the two modes is not zero, as shown by the red line. As the resonant frequencies of the two modes are equal, a clear standard transparent window will be displayed, as shown by the blue line. From the slope curves presented in Figure 3f, the maximum slope of the transparent window is nearly equal to the maximum slope of the Fano resonance, but it is worth noting that the visually small Lorentz representation of the high-*Q* mode in the spectrum is not conducive to observing slope changes using Fano resonance. This can be reflected in Figure 3a,b. In addition, the maximum slope of the transparent window is 8.3 times greater than the maximum slope of the Lorentz of the low-*Q* mode. In other words, utilizing the slope of the transparent window for sensing has a detection sensitivity 8.3 times higher than that of the low-*Q* mode.

## 3. Experiments and Discussions

### 3.1. Regulation of the EIT-Like Effect

Mode-coupling and EIT-like effect can generate between abundant modes when a tuning method is applied to the microresonator, including thermal tuning [9,27], pressure-induced tuning [28], field manipulation [29], etc. Similarly, external coupling losses $\kappa_{A/i}$ and frequency zones $\Delta \omega_{A/i}$ can be adjusted to realize the EIT-like and Fano effects by changing the coupling state between the tapered fiber and the microresonator [30,31,32]. Experimentally, we can obtain tunable coupling losses $\kappa_{A/i}$ and frequency zones $\Delta \omega_{A/i}$ by moving the SNAP microresonator vertically along its longitudinal axis.

Two resonant modes with close resonant wavelength points were selected to monitor mode-coupling; resonant spectra were obtained at ∆z = 5 μm, 10 μm, 15 μm, 20 μm, and 25 μm and are depicted in Figure 4a–e. Here, ∆z is the axial moving distance of the SNAP microresonator. Dual-mode-coupling occurs between Mode-1 (low *Q* mode) and Mode-2 (high *Q* mode), which generates the Fano lineshapes and EIT-like effects, subsequently converted to the standard transparent window when the resonant frequencies of the two modes coincide. Figure 4f–j are the fitting curves corresponding to (a–e). Fano resonance phase varies due to the long narrow mode field properties of the SNAP microresonator. For the coupled-mode induced transparency, Mode-2 should be in the over-coupled state ($\kappa_{2}>\gamma_{2}$), but Mode-1 should be in the under-coupled state ($\kappa_{1}<\gamma_{1}$) or nearly critical coupled region ($\kappa_{1}\sim \gamma_{1}$). In Figure 4d,e, Mode-3 approaches Mode-2 as the tuning operation is in progress. This reveals the implementation of multi-mode-coupling.

We further investigated the multi-mode-coupling by tuning three resonant modes. The transmission spectra were obtained at ∆z = 23 μm, 24 μm, 25 μm, 26 μm, 27 μm, and 28 μm, as shown in Figure 5a–f. Mode-1 and Mode-2 were tuned to the standard transparent window first due to their close proximity. Fano resonance can be observed resulting from the interference of Mode-3 with Mode-1, as shown in Figure 5a–e. During the tuning process, the motion of Mode-3 again demonstrates the implementation of DEIT-like effects according to multi-mode-coupling theory. When Mode-3 is very close to Modes 1/2, as shown in Figure 5f, a clear upward peak appears at the position of Mode-3, indicating the implementation of multi-transparent windows [27,33]. For the single SNAP microresonator, multi-mode-coupling is verified simply by vertically changing the coupling position along the longitudinal axis of the microresonator.

### 3.2. Automatic Regulation of the EIT-Like Effect

The SNAP microresonator featured a distinctive axial mode field distribution. This distinct field distribution served as a robust foundation for the axial separation sensing capabilities. We defined the coupling position of the microresonator–taper coupling system as the coupling region, and the area below the coupling position with distance L as the sensing region. The change in effective refractive index on the sensing surface within the mode volume of the microresonator is the primary trigger for frequency shifts [34,35]. Due to the extremely small curvature and the long narrow axial mode field distribution of the SNAP microresonator, the effective mode field length can reach several hundred micrometers. The axial separation sensing system demonstrated in the experiment is shown in Figure 6a. The tunable laser (TL, narrow-linewidth laser, 1550 nm) is coupled into the tapered fiber waveguide and collected by the photodetector (PD). Photons that match the resonant frequency will exhibit Lorentz dips in the spectrum and be displayed on the oscilloscope (OSC). The polarization state of light is adjusted by the polarization controller (PC). The function signal generator (FSG) is used to scan the laser wavelength. In order to achieve real-time and fast spectral acquisition, we connect a data acquisition card (DAC) at one end of the fiber-optic splitter (FOS). The fiber-optic attenuator (FOA) is used to adjust the optical power. One stem of the SNAP microresonator is fixed on an iron pedestal using a quartz tube; the other stem serves as the sensing region and ready to come into contact with analyte. Deionized water is carried by a microfluidic channel and placed on the displacement table (DPT). The height of the DPT can be precisely adjusted to bring the sensing region into contact with the aqueous environment. Experimentally, L was set to 300 μm.

Figure 6b depicts a detailed illustration for the axial separation sensing mechanism. The analyte in liquid interacts with the mode field far from the coupling plane, inducing a shift or deformation of the resonance spectrum. The analyte concentration and activity level can be effectively calculated by precisely measuring these variations. Figure 6c is the physical device corresponding to Figure 6a. The sensing region of the SNAP microresonator is immersed in deionized water while the spectra are monitored in real-time. It is interesting that an automatically adjustable EIT-like effect is discovered, as shown in Figure 6d. In this case, the sensing region of the SNAP microresonator is always immersed in liquid and does not require any additional operation. In fact, the microfluidics effect of liquid induces a continuous change in effective refractive index per unit area. The liquid microfluidic effect here can be understood as the liquid constantly flowing, which results in effective refractive index fluctuations at the boundary between the wall and the liquid. In addition, different resonant modes have different tuning rates. For this effective refractive index gradient, there will be automatic mode-coupling and tunable transparent window without human intervention. The fitting of EIT-like effect is shown in Figure 6e.

### 3.3. Liquid Refractive Index Sensing

A simulation model can be used to characterize the resonant frequency shift induced by variations in liquid refractive index. The simulation model is established in COMSOL. We simulated the change in liquid refractive index by increasing the air refractive index on the surface of the microresonator, i.e., n increased from 1.000 to 1.008. This rational substitution is based on two facts: a portion of the WGM field energy emits evanescently into air/liquid; the change in air/liquid refractive index will induce a change in effective refractive index between the wall and the detection environment. The cross-sectional WGM distribution corresponding to each air refractive index value is shown in Figure 7a. Simultaneously, we plotted the transmission spectra as shown in Figure 7b. With increasing n, the resonant wavelength shifts towards longer wavelengths. As n increases further, WGMs are strongly absorbed by the environment, leading to a decrease in electric field intensity, as shown in Figure 7c. Experimentally, we injected glucose solutions with varying concentrations into microfluidic channel, and the liquid refractive indices were measured as 1.3335 (zero concentration), 1.337, 1.341, 1.344, 1.3475, and 1.3505, respectively. The sensing region of the SNAP microresonator was immersed in liquid and L was adjusted as 300 μm. As shown in Figure 7d, sharp transparent windows can be used to observe the resonant wavelength shift with varying liquid refractive indices. These transmission spectra were extracted from the automatically adjustable EIT-like effects corresponding to each liquid injection stage, and exhibited the characteristics of the standard transparent window. The sharp transparent window provides great convenience in observing the resonant wavelength shifts. The fitting curves supporting these transparent windows are shown in the inset of Figure 7d. The resonant wavelength underwent a red shift from 1550.078 nm to 1550.113 nm. The resonant wavelength shifted as a function of the liquid refractive index as shown in Figure 7e. The refractive index sensitivity (RIS) was calibrated as 2058.8 pm/RIU. It is worth noting that the axial separation sensing we proposed has been successfully implemented in an aqueous environment.

### 3.4. Liquid Displacement Detection

The slope of the transparent window can be altered by adjusting the relevant parameters of the high *Q* (low *Q*) mode. The high-slope variation of the transparent window at a fixed wavelength is superior to mode shift, regarding sensing performance. Experimentally, we shortened the distance L between the coupling region and the liquid surface by adjusting the DPT. This process results in an increase in effective action area between WGMs and testing liquid. In the meantime, a corresponding variable slope of the transparent window can be observed. The mechanism is better explored by building the electric field distribution model in Figure 8a. The curvature of the SNAP microresonator in the model is 0.0119 μm^−1^, the wall thickness is 2 μm, the maximum radius is 7.2 μm, and one stem of it is immersed in liquid. The refractive indices of air and liquid are 1.0 and 1.3335, respectively. The liquid surfaces are set to −10 μm, −5 μm, and 0 μm. The axial WGMs with *q* = 6 are excited (the center frequency of pumped laser is 300 THz), as shown in (i), (ii), and (iii) of Figure 8a. The electric field energy outside the microresonator interacts with liquid as evanescent waves, and thus induces visual spectra deformation or shift. From the radial section line direction, as shown in Figure 8(a-iv), the radial evanescent field energy emitted in liquid is greater than that in air, indicating that the SNAP microresonators are suitable for detecting aqueous environments. As the liquid surface height is lifted, judging from the axial section line direction, the axial evanescent field distribution is more in contact with liquid, and consequently results in a variation in resonant frequency response, as shown in (i), (ii), and (iii) of Figure 8b. Experimentally, we set the liquid lifting height as 0 μm, 100 μm, and 200 μm, respectively. For each liquid lifting height, we selected one representative transmission spectrum, as shown in Figure 8c, which is depicted in red (0 μm), green (100 μm), and blue (200 μm), respectively. The liquid lifting operation resulted in an increased peak value of the transparent window, and further increased the slope. Figure 8d is the localized magnification for the spectra in (c). The slope of the transparent window was monitored at each height, as shown in Figure 8e. At the three moving distances (MD, 0 μm, 100 μm, 200 μm) of the DPT, the slope data are divided with three different colors from left to right. The mean values of the slopes at the three heights are 149.5 ± 17.2 V/nm, 197.6 ± 23.5 V/nm, and 377.8 ± 37.7 V/nm, respectively. The mean value of the slopes as a function of the moving distance is displayed in the inset of Figure 8e. The mean value increases exponentially with an increase in MD, which is compatible with the increase in effective cone angle of the cylinder microcavity discussed in the literature [36,37]. Essentially, it will lead to the axial mode field being localized as the liquid approaches the coupling plane, which in turn will contribute to the resonant spectra showing sharper features. The probability distribution histogram of the slope values is shown in Figure 8f. The data distribution shifts to higher values with increasing MD, and the probability distribution curve basically shows a normal distribution. The high slope of the transparent window can amplify weak sensing signals. Based on the EIT-like effect and high slope of the transparent window, SNAP microresonators can be considered as ultra-high sensitivity displacement detectors [38,39,40]. Furthermore, the axial separation sensing method supported by the SNAP microresonator avoids perturbation of the WGM field by the detection environment, prolonging the lifespan of the tapered fiber and ensuring the stability of the detection signal.

## 4. Conclusions

In summary, we achieved regulatable EIT-like and double EIT-like effects on a single SNAP microresonator by changing the coupling position of the microresonator–taper coupling system. In addition, an axial separation detection can be implemented on the SNAP microresonator. On this basis, a liquid refractive index sensitivity of up to 2058.8 pm/RIU achieved by utilizing the sharp lineshape of the transparent window. It is interesting that the refractive index gradient effect caused by liquid microfluidics results in an automatically adjustable EIT-like effect. By shortening the distance between the coupling region of the SNAP microresonator and the liquid surface, we observed a corresponding change in the slope of the transparent window. The high slope spectrum of the transparent window makes it easy to observe a weak sensing signal at a fixed wavelength. Based on this phenomenon, SNAP microresonators can be considered as ultra-high sensitivity displacement detectors. The EIT-like effect implemented on a single SNAP microresonator and the axial separation sensing method utilizing the sharp lineshape of the transparent window make microresonators very suitable for optical switches, multi-channel optical communication technology, and ultra-high sensitivity biochemical sensing.

## Figures and Tables

**Figure 1 sensors-24-07069-f001:**
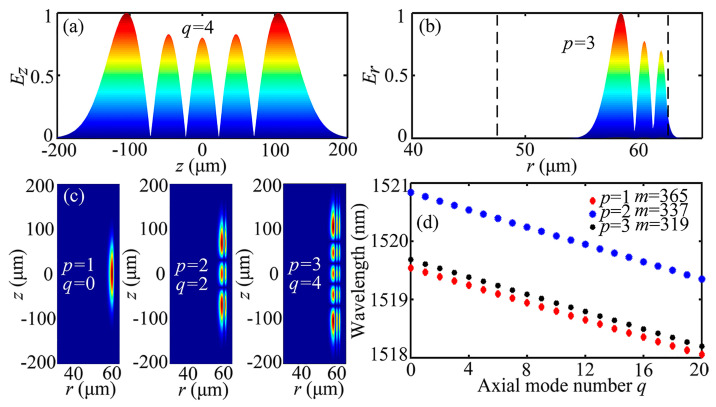
(**a**) Normalized electric field distribution for the fifth order axial mode. (**b**) Normalized electric field distribution for the third order radial mode. (**c**) Cross-sectional views of the normalized electric field distributions with (*p* = 1, *m* = 365, *q* = 0), (*p* = 2, *q* = 2, *m* = 337), and (*p* = 3, *m* = 319, *q* = 4). (**d**) Resonant wavelengths as a function of axial mode numbers *q* with measurement size.

**Figure 2 sensors-24-07069-f002:**
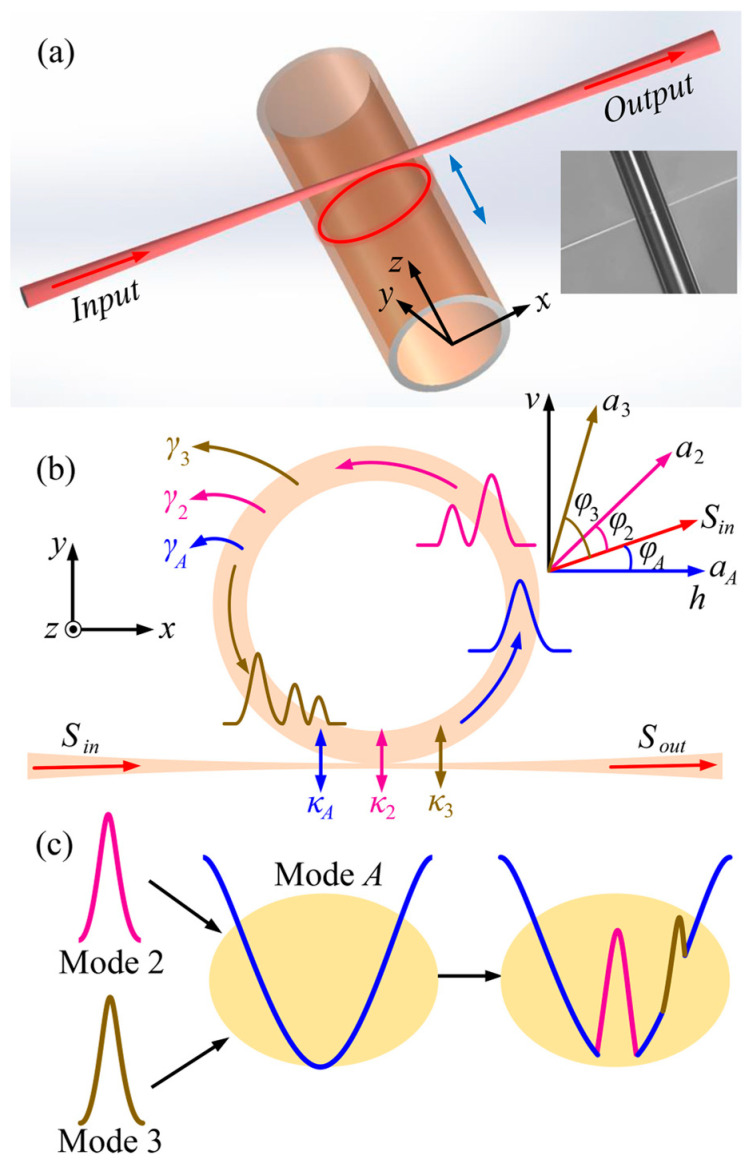
(**a**) Schematic diagram for the microresonator–taper coupling system. (**b**) Schematic diagram for the three-mode-coupling microcapillary system. (**c**) Illustration for the three-pathway interference effect induced by the three kinds of WGMs.

**Figure 3 sensors-24-07069-f003:**
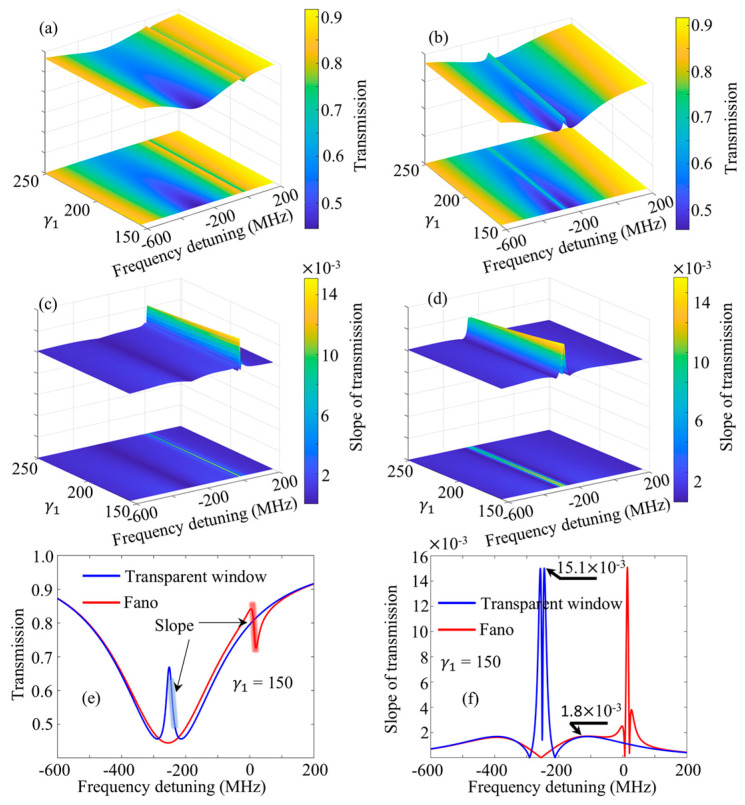
(**a**) Simulation spectrum varies with $\gamma_{1}$. The resonant frequency difference between high-*Q* mode and low-*Q* mode is not zero and induces the Fano resonance. (**b**) Transparent window spectrum varies with $\gamma_{1}$. The resonant frequency difference between the two modes is zero and induces the standard transparent window. (**c**,**d**) show the derivatives of the spectra with respect to ω in (**a**,**b**), respectively. (**e**) Transparent window lineshape (blue line) and Fano lineshape (red line) with $\gamma_{1}$ = 150. (**f**) Slope curves of transparent window and Fano resonance in case of (**e**).

**Figure 4 sensors-24-07069-f004:**
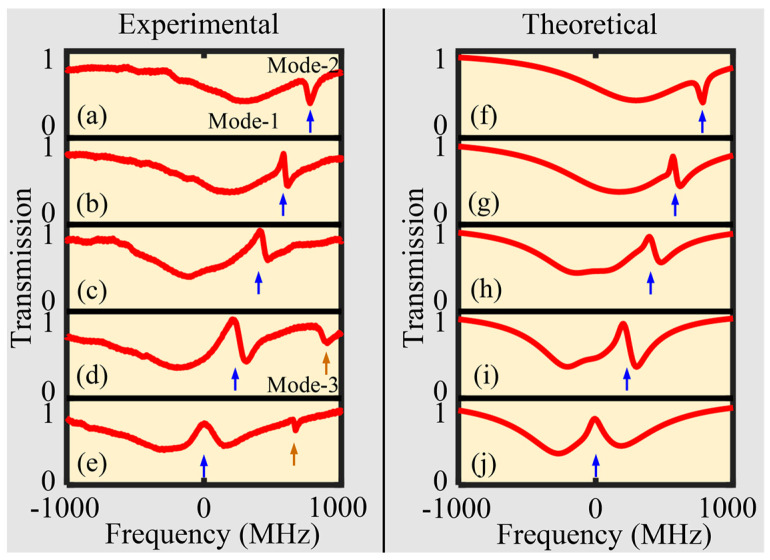
Dual-mode-coupling and the corresponding Fano/EIT–like effects realized in the SNAP microresonator. (**a**–**e**) Experimental normalized transmission spectra at different coupling positions, ∆z = 5 μm, 10 μm, 15 μm, 20 μm, and 25 μm. (**f**–**j**) Fitting experimental data corresponding to (**a**–**e**). The simulation data are set to: $\left[\gamma_{1},\gamma_{2},\kappa_{1},\kappa_{2},∆\omega_{1},∆\omega_{2},\varphi_{1},\varphi_{2}\right]$ = [783.2, 26.4, 183.2, 12.8, 295, 772.7, 0.28, 8.9], [643.2, 1, 263.2, 74.8, 195, 564.7, 0.52, 8.76], [373.2, 5, 263.2, 124.8, −15, 394.7, 0.73, 8.91], [293.2, 10, 240.2, 140.8, −75, 204.7, 0.71, 9], [153.2, 10, 290.2, 130.8, −100, −19.7, 0.69, 12.25].

**Figure 5 sensors-24-07069-f005:**
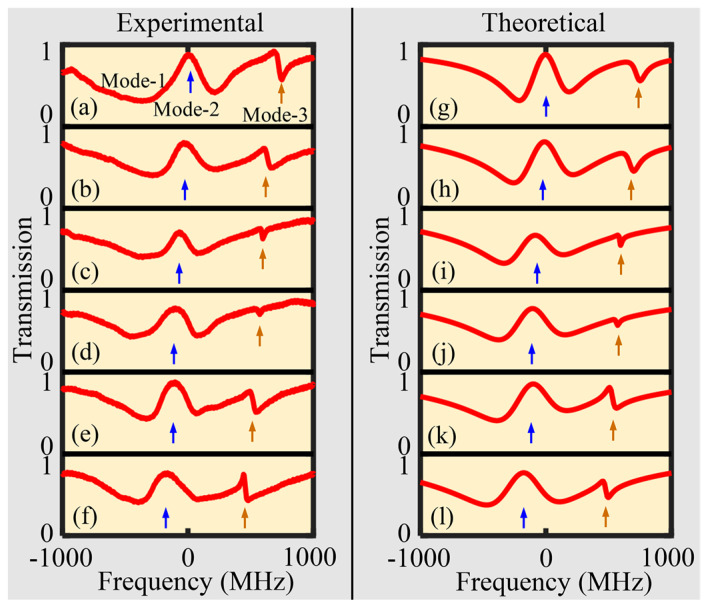
Multi-mode-coupling and DEIT effect realized in the SNAP microresonator. (**a**–**f**) Experimental normalized transmission spectra at different coupling positions. (**g**–**l**) Theoretical transmission lineshapes for fitting experimental data. The simulation data are set to: $\left[\gamma_{1},\gamma_{2},{\gamma_{3},\kappa }_{1},\kappa_{2},\kappa_{3},{∆\omega }_{1},∆\omega_{2},∆\omega_{3},\varphi_{1},\;\varphi_{2},\;\varphi_{3}\right]$ = [403.2, 576.4, 77.4, −3403.2, −290.8, −12, 170, −5, 740.7, 0.17, 8.82, 8.6], [1253.2, 585.4, 77.4, −3203.2, −287.8, −12, 60, −15, 690.7, 0.17, 8.82, 8.7], [1753.2, 625.4, 22.4, −4003.2, −287.8, −1, 20, −90, 590.7, 0.12, 8.82, 9.6], [1753.2, 625.4, 32.4, −4003.2, −297.8, −1, −50, −110, 565.7], [1753.2, 625.4, 32.4, −4003.2, −304.8, −5, −50, −110, 535.7, 0.12, 8.82, 8.4], [1753.2, 625.4, 32.4, −3503.2, −294.8, −5, −180, −180, 485.7, 0.12, 8.82, 8.7].

**Figure 6 sensors-24-07069-f006:**
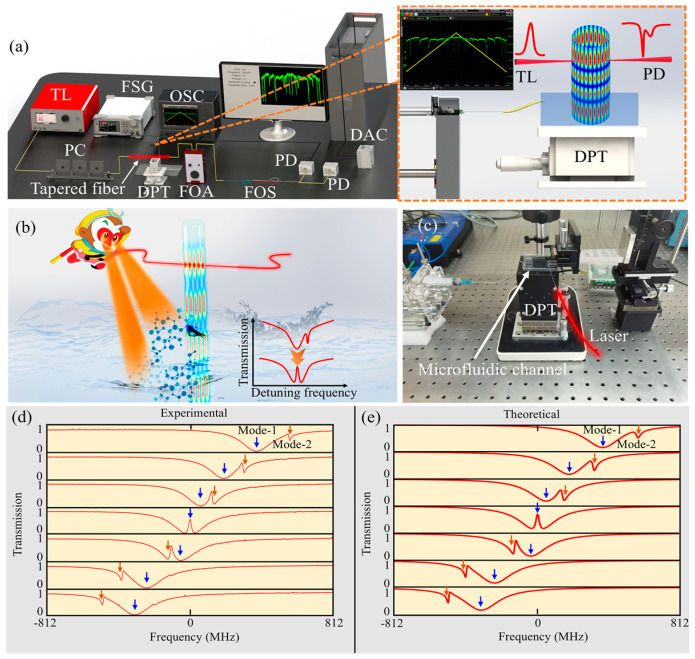
(**a**) Experimental apparatus for the axial separation sensing. (**b**) Schematic diagram of the axial separation sensing technology for detecting the aqueous environment. (**c**) The physical device corresponding to (**a**). (**d**) Automatically adjustable EIT-like effect when the sensing region is immersed in deionized water. (**e**) Theoretical transmission lineshapes for fitting experimental data. The simulation data are set to: $\left[\gamma_{1},\gamma_{2},{\;\kappa }_{1},\kappa_{2},{∆\omega }_{1},∆\omega_{2},\varphi_{1},\;\varphi_{2}\right]$ = [94.2, 11.3, 31.4, 0.063, 375, 562, 0.24, 8.78], [94.2, 7.54, 31.4, 0.063, 187, 312, 0.24, 8.85], [94.2, 6.91, 31.4, 0.063, 62.5, 137.4, 0.44, 8.91], [94.2, 3.14, 31.4, 0, 0, 0, 0.58, 9.23], [94.2, 3.14, 31.4, 0, −50, −125, 0.52, 10.88], [94.2, 3.14, 31.4, 0, −250, −400, 0.61, 11.91], [94.2, 3.14, 31.4, 0, −324.6, −500, 0.61, 11.97].

**Figure 7 sensors-24-07069-f007:**
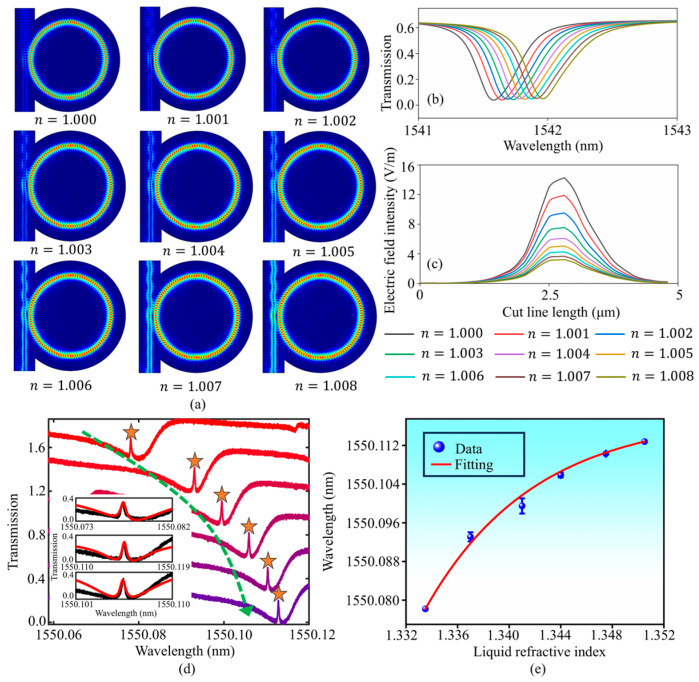
(**a**) Cross-sectional WGMs distribution with increasing air refractive indices. (**b**) The resonant wavelength shifts towards longer wavelengths as n continues to increase. (**c**) The electric field intensity distribution decreases as n continues to increase. (**d**) Evolution of the transmission spectrum with increasing liquid refractive index. The green arrow indicates the wavelength shift direction. (**e**) The resonant wavelength shifts as a function of the liquid refractive index.

**Figure 8 sensors-24-07069-f008:**
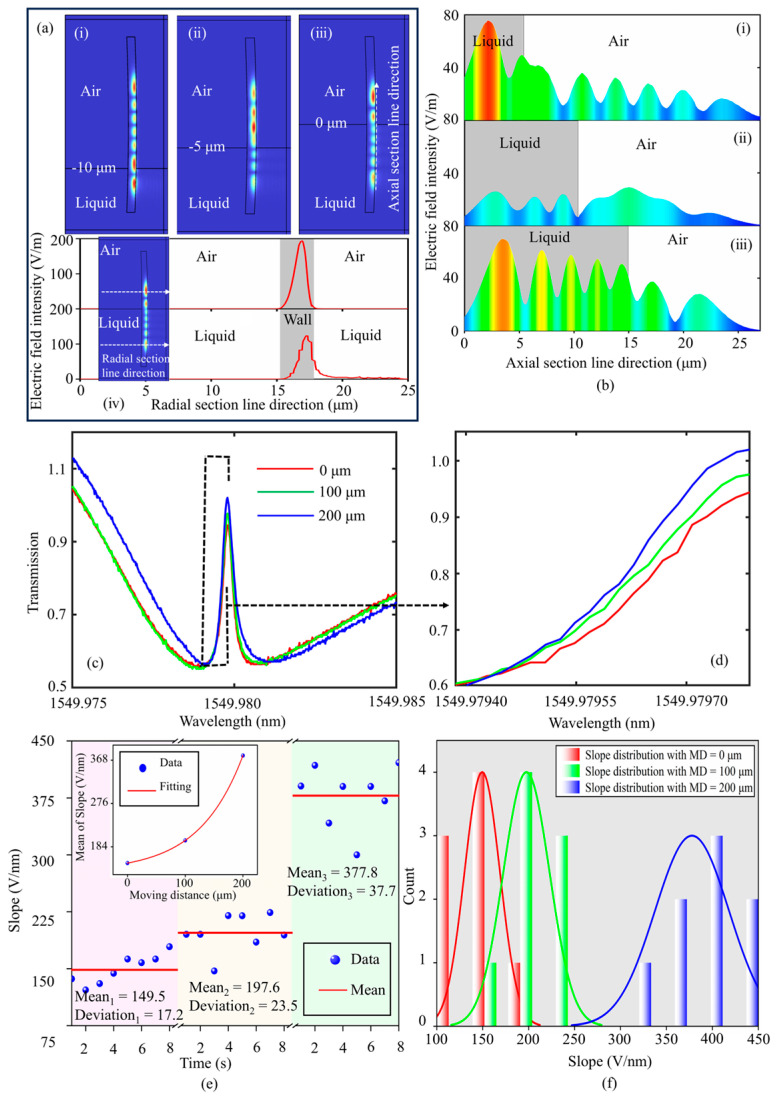
(**a**) Simulation model for the electric field distribution. One stem of the microresonator is immersed in liquid. Axial mode with *q* = 6 is excited in the model. The liquid surfaces are set to −10 μm, −5 μm, and 0 μm; the corresponding electric field distribution models are shown in (**i**), (**ii**), and (**iii**), respectively. The electric field intensity in liquid and air along the radial section line direction is shown in (**iv**). (**b**) Axial electric field intensity distribution with different liquid surfaces along the axial section line direction. (**i**), (**ii**), and (**iii**) are electric field distribution models as the interaction area between the liquid and the sensing region of the SNAP microresonator is increased. (**c**) Representative transmission spectra corresponding to each liquid lifting height; red (0 μm), green (100 μm), and blue (200 μm). (**d**) Localized magnification for the spectra in (**c**). (**e**) The slope values of the transparent window with MD = 0 μm, 100 μm, 200 μm, respectively. The slope data are divided with three different color intervals from left to right, corresponding to each MD. The mean values of the slopes as a function of the MD are displayed in the inset. (**f**) Probability distribution histogram of the slope values corresponding to various MD.

## Data Availability

Data will be made available on request.

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
