# Peer review of "Regulation and Liquid Sensing of Electromagnetically Induced Transparency-like Phenomena Implemented in a SNAP Microresonator"

_sensors, 2024, doi:10.3390/s24217069_

Round 1
Reviewer 1 Report
Comments and Suggestions for Authors
Dear Prof.
A good work is introduced addresses (beginning from the title) the electromagnetically induced transparency (EIT). It is a good work. But, no EIT is seen. There is an electromagnetically induced dip with single peak not EIT. So how one call it EIT. In this case substantial changes must be done overall the work. The main point is what is addressed here. In addition, the following points must be addressed,
1- Lines 81-82: “Electric field distribution can be obtained by substituting these parameters into the simulation model”. Please specify the simulation model. If you use a package tell what it is.
2- L 93-95: “It can be seen that, for example, the resonant wavelength of mode (3, 319, q) is localized between modes (1, 365, q) and (2, 337, q). The wavelength point of modes (1, 365, q) and (3, 319, q) is closely adjacent at the same axial mode”. Fig. 1(d): why the p=1,3 modes are adjacent while p=2 is not between them and far. Discuss this, please.
3- L 114-117: For the coupled mode Eqs. (1), (2), write their origin. For example Maxwell Eq. Really, they like those used to describe fields in the laser with feedback. I think, explanation of the system with the associated Eqs. (3)-(5) is confidential.
4- Please explain how use the model above to get Fig. 3. Should you solve it analytically, or numerically. specify and what are the boundary conditions used as you plot between T and in Eqs. (1), (2). You put
value what is for other values
. detailed explanation is required.
5- L 61-62: “the slope of the transparent window for sensing has a detection sensitivity 8.3 times higher than that of the low-Q mode. Figures 3(c) and (d) show the derivatives of the spectra with respect to w in Figures 3(a) and (b), respectively”. But in c and d the curves are ~ the same and this 8.3 times higher is not shown? Why?
6- L 170: . What is the unit? Is is normalized? To what?
7- L 176: use the following ref. after ref. [25]:
B. Al-Nashy, S. M. M. Amin and Amin H. Al-Khursan,"Kerr effect in Y- configuration double quantum dot System", J. Opt. Soc. Am. B Vol. 31, (2014) 1991-1996.
8- L 197: “in Figures 4(d)-(e), the Mode-3 approaches Mode-1”. Im not see such approaching. The last arrow in Fig. 4(d) approaches mode-2. Please check or answer. Fig. 5 exactly shows what I say.
9- L 213: “DEIT effect was realized, illustrated in Figure 5(f)”. fig 5. From a-f are the same except c, d. So a,b,e are also DEIT. Specify.
10- L 256: “shown in Figure 6(f)”. no “f” is it Fig. (d). check.
11- L 273: “increasing the air refractive index on the surface of the microresonator, that is, n increases from 1 to 1.008”. Please how it increased in practice. If this is an approx., is it accepted, why?
12- How Fig. 7 is plotted. Write the codes or software used in each case with abbreviated description.
13- L 288: “EIT-like effect to create a transparent window”. Im not see the EIT effect nor its trans. Window. EIT require two peaks with window between them.

Reviewer 2 Report
Comments and Suggestions for Authors
Optical microcavity plays an important role in quantum optics, quantum information and quantum sensing etc. The authors investigated the tunable electromagnetically induced transparency (EIT) effects in a taper fiber-microresonator coupled system. By using the surface nanoscale axial photonics (SNAP) microresonator, they experimentally investigated the single-EIT and double-EIT (DEIT) effects with tunable resonant frequencies. Moreover, based on the sharp lineshape of the EIT effects and the Fano resonance, they achieved high-precision sensing of liquid refractive index and displacement in experiments. This work may further promote the development of fields such as optical sensing and optical information processing. In addition, this manuscript is also well written in English and well-organized. Therefore, I happily recommend its publication in Sensors.
Reviewer 3 Report
Comments and Suggestions for Authors
The authors fabricated SNAP microresonators and tested their applicability for refractometric and absorption sensing by monitoring WGM shift and shape. Because of the great density of modes in their resonators they monitor the EIT and Fano lineshapes resulting in interaction of high Q and low Q modes. The refractometric measurements in standard glucose solutions give about 2 nm/RIU which is not very much (it's less than usually obtained in microspheres, microrings and other WGM), but it's something. They applied the absorption sensing mechanism to the measurement of resonator dipping depth in water and obtained a measurable response, although slightly unstable. The research presented is of interest to the scientific community and is suitable for Sensors. There are a few issues:
1) In Figures 4 and 5 the values of fitting parameters are given without dimensions. These need to be specified. The Ï• values are (I assume) in radians, but the γ and κ and Δω values should have dimensions of s-1. Form Figures 4 and 5 I assume the values given are in MHz, but the authors should specify that somewhere in the text.
2) In Figure 4 all 8 and in Figure 5 all 12 fitting parameters are varied. Why? In the text it correctly says that the change of the coupling point changes the values of κ and Δω. However (since it is the same modes they are monitoring) the values of γ should be the same. The change of coupling point doesn't change the intrinsic Q-factor of a given mode. In fact the authors sometimes keep the γ values constant (for example in Fig 5 (i),(j),(k) and (l) γ1=1753.2) but then sometimes they inexplicably change them (in Fig 5 (g) γ1=403.2). For the same mode the γ value should be kept constant for all fits.
3) It is not clear what is exactly the experimental measurement in Figure 6(d). Are different presented waveforms just different time periods after immersing the resonator in water? If so then what is the time scale? Or are they different dipping depths?
4) In the paper the authors claim (for example on line 257) that the different modes have different refractometric sensitivities (which is generally to be expected). However, in Fig. 7(d) it seems that the shift of the EIT feature is the same as the shift of the low Q mode. Is this a feature that is present in all the measurements or is it just a coincidence that in Fig(d) the low Q and high Q modes have almost the same sensitivities.
5) In Figure 8(c) and (d) and on line 342 the slope is given in units of V/m. Is this a mistake? It seems to me from Fig 3 that the slope should be in MHz-1. Also, since the values of the slope are calculated as a mean of a series of measurements, the standard deviation should be also included (slope=mean±deviation).
6) It would be nice if the authors could include in Figure 8 the actual waveforms form which the slopes in Fig8(c) are calculated. Of course not all 24 measurements, but at least of one representative measurement from the three regions (0, 100 and 200 µm dipping depth)
Comments on the Quality of English LanguageThe English of the paper is not very good. Overall the paper is more or less understandable, but the low quality of English makes it difficult to read. I know that a certain amount of broken English is usually tolerated in MDPI journal but in this case the quality of English should really be improved, so I would suggest for the authors to find somebody fluent in English to correct their paper.
Round 2
Reviewer 1 Report
Comments and Suggestions for Authors
ok